# Is Cancer Metabolism an Atavism?

**DOI:** 10.3390/cancers16132415

**Published:** 2024-06-29

**Authors:** Eric Fanchon, Angélique Stéphanou

**Affiliations:** Université Grenoble Alpes, CNRS, UMR 5525, VetAgro Sup, Grenoble INP, TIMC, 38000 Grenoble, France

**Keywords:** cancer theory, gene expression, hypoxia, metabolic plasticity, Warburg effect

## Abstract

**Simple Summary:**

The atavistic theory of cancer suggests that cancer development proceeds through cells reverting to ancient survival mechanisms. The Serial Atavism Model (SAM) expands on this, proposing that cancer progresses through multiple stages of reversion to earlier evolutionary forms with cells losing modern traits and regaining primitive ones. One example is the Warburg effect, where cancer cells prefer a type of energy production used by ancient cells before Earth’s atmosphere had oxygen. However, this review argues that cancer metabolism is too complex to be fully explained by this theory. Cancer cells exhibit a wide range of metabolic behaviors that do not fit neatly into a pattern of reverting to an ancient state, indicating that the SAM may not provide a complete understanding of cancer.

**Abstract:**

The atavistic theory of cancer posits that cancer emerges and progresses through the reversion of cellular phenotypes to more ancestral types with genomic and epigenetic changes deactivating recently evolved genetic modules and activating ancient survival mechanisms. This theory aims at explaining the known cancer hallmarks and the paradox of cancer’s predictable progression despite the randomness of genetic mutations. Lineweaver and colleagues recently proposed the Serial Atavism Model (SAM), an enhanced version of the atavistic theory, which suggests that cancer progression involves multiple atavistic reversions where cells regress through evolutionary stages, losing recently evolved traits first and reactivating primitive ones later. The Warburg effect, where cancer cells upregulate glycolysis and lactate production in the presence of oxygen instead of using oxidative phosphorylation, is one of the key feature of the SAM. It is associated with the metabolism of ancient cells living on Earth before the oxygenation of the atmosphere. This review addresses the question of whether cancer metabolism can be considered as an atavistic reversion. By analyzing several known characteristics of cancer metabolism, we reach the conclusion that this version of the atavistic theory does not provide an adequate conceptual frame for cancer research. Cancer metabolism spans a whole spectrum of metabolic states which cannot be fully explained by a sequential reversion to an ancient state. Moreover, we interrogate the nature of cancer metabolism and discuss its characteristics within the framework of the SAM.

## 1. Introduction

Cancer is one of the most intensely studied biological phenomena, and yet it remains poorly understood. Despite many advances, patient survival has not changed over several decades of research for many cancers. One observation is that this lack of significant progress is attributable to a misunderstanding of the fundamental nature of cancer. Several initiatives were launched to rethink the conceptual foundations of cancer, one by convening physicists to the debate to provide a new vision [1]. It is in this context that the “atavist theory of cancer” (re)emerged, which led to a radical paradigm shift opening the path to a new way of thinking about therapy [2,3,4].

The idea that cancer appears as a resurgence of ancient traits from the evolutionary past of the cell was already postulated by several authors, the first being Snow in 1893 [5], then Boveri in 1914 [6], Roberts in 1926 [7] or more recently Israel in 1996 [8]. The atavistic theory, formulated in 2011 by Davies and Lineweaver [1] and Vincent [9], proposes that the emergence and development of cancer is explained by the regression of cellular phenotypes toward primitive types, featuring progressive deactivation and alteration at the genomic and epigenetic level of biological systems recently acquired in evolution in favor of ancestral biological systems giving cells a greater ability to survive altered and unstable environments. This theory, supported by phylostratigraphic observations, proposes a simpler mechanism to the commonly accepted one of purely random mutation.

In Mark Vincent’s version of the theory, cancer progression is the result of the unfolding of “a highly conserved survival program, honed by the exigencies of the Pre-Cambrian” [9]. This program is reactivated by an accidental event (stress, gene mutation or epimutation), but once it is triggered, its unfolding is deterministic. At that time, cells only existed individually and had to survive in harsh environmental conditions dominated by intense radiation and the almost total absence of oxygen in the primitive atmosphere (Figure 1). The change in environmental conditions and in particular the oxygenation of the atmosphere gradually led cells to cooperate for their survival. One of the key events was that the cells have integrated by endosymbiosis a proto-bacteria—ancestor of the mitochondria [10]—which allows the detoxification of oxygen, providing a defensive advantage to the cell in the face of oxygen attacks. Another key factor in this process was the synthesis of molecules such as collagen—which requires the presence of oxygen—which provided the glue (i.e., extracellular matrix) necessary for cells to develop a communication [11]. Over the course of evolution, cells have acquired characteristics allowing them to form multicellular organisms [12,13] and have neutralized and/or deactivated the primitive characteristics of individual survival [14]. They have acquired a “social” behavior favoring collective survival and cooperation.

From this new perspective, the state of cancer is no longer seen as the result of random mutations leading to a deregulated behavior but as a state of absolute survival pre-existing at the heart of each cell and reactivated under conditions of stress [1,8]. This vision allows explaining why cancer affects all forms of animal life and why cancer development is so predictable and “obeys” predetermined patterns (the same ones used to describe the evolutionary stage, i.e., grading of cancer) [15]. On the other hand, this vision imposes that the emergence of cancer is primarily the reactivation of the expression of dormant genes under certain environmental stresses rather than due to random mutations which cannot explain the convergence toward the well-characterized (predetermined) cancerous state [16]. Genetic instability is of course involved but at a later stage. It is explained by the deactivation of the systems of repair [9].

Since the atavistic theory was formulated in 2011 [1,9], it has for many years remained quite confidential in the biomedical community, which either (i) has not been aware of it (publications in less visible journals by this community) (ii) or has not appreciated its potential importance for progress in understanding cancer and how to treat it. Some works tend to show that the layers of cooperation are intertwined and follow a well-defined order. Characters are thus lost sequentially—respecting an apparent hierarchy—with the loss of control of proliferation first and genetic instability occurring later [15]. The importance of epigenetics in the phenotypic expression of cells is now well confirmed [17].

One of the key drivers of reversion is oxygen deprivation. The cellular response to hypoxia is a change in energy metabolism from respiration (aerobic oxidative phosphorylation via mitochondria) to fermentation (anaerobic glycolysis). A particularity of tumor cells is that they tend to favor fermentation, independently of the level of oxygen, that can be interpreted as a return to the ancestral metabolic pathway. This feature is known in the literature as the Warburg effect [18,19].

**Figure 1 cancers-16-02415-f001:**
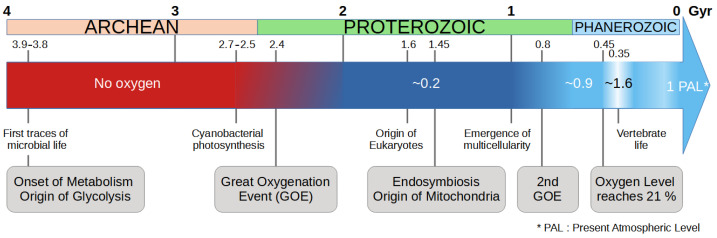
Timeline of the evolution of life in relation to the presence of oxygen in the atmosphere [20]. Key events for the evolution of the cell energy metabolism are given in the gray boxes.

## 2. The What and the Why of the Atavistic Theory of Cancer

The atavistic theory of cancer has been proposed, in part, to explain the remarkably consistent and predictable pattern of cancer progression despite the randomness of genetic mutations that drive the disease. This consistency is surprising, because one might expect the random nature of mutations to result in a wide variety of cancer behaviors and states. Cancer progression typically follows a well-defined series of stages, from benign hyperplasia to increasingly aggressive forms like dysplasia, and ultimately invasive and metastatic cancer. Despite genetic diversity and the randomness of mutations in cancer cells, tumors tend to converge on similar phenotypic traits and behaviors, such as sustained proliferation, resistance to apoptosis, and metastasis capability, suggesting an underlying common pathway or set of pathways. The integrated view of an “Oncospace” was recently proposed to visualize the finite number of observable cancer states. These are contained in a three-dimensional space that combines evolutionary (genome instability), ecological and developmental alterations [21] that constrain cancer trajectories. The atavistic theory, on the other hand, postulates that cancer cells reactivate ancient genetic programs from unicellular ancestors, which are robust and well conserved due to their essential survival and reproductive roles in early life forms. This reactivation results in a simplified evolutionary pathway, as cells revert to these ancient survival mechanisms, leading to a streamlined and consistent evolution toward similar stages of aggressiveness across different cancers. According to the theory, cancer cells progressively lose the functions of differentiated, specialized cells and adopt traits of early life forms, explaining the similar aggressive behaviors of advanced cancers from different tissues [3]. The hallmarks of cancer, such as sustained proliferative signaling, the evasion of growth suppressors, resistance to cell death, and the ability to invade tissues and metastasize [22], are seen as expressions of these ancient survival strategies. This commonality supports the idea that cancer progression follows a predictable path, tapping into a universal, ancient toolkit. Consequently, the mutations and epigenetic changes observed in cancer cells are biased toward disrupting the more recently evolved, less stable genes while preserving the ancient, core survival functions. Hence, the atavistic theory of cancer helps explain the paradox of cancer’s predictable progression despite the underlying randomness of genetic mutations [23].

## 3. Update of the Theory

The Serial Atavism Model (SAM), proposed by Lineweaver and colleagues in 2021 [23], is an enhanced version of the atavistic theory of cancer that provides a more detailed understanding of cancer progression. Unlike the original atavistic model, which suggested a single reversion from multicellularity to unicellularity, the SAM posits that cancer progression involves a series of atavistic reversions. Each reversion represents a step back to an earlier evolutionary state, where cells increasingly exhibit traits of their ancestors. The SAM proposes that as cancer develops, cells undergo a stepwise regression through various evolutionary stages, with more recently evolved traits being lost first, which is followed by the reactivation of older, more primitive traits. The model postulates that the sequence of these atavistic reversions correlates with the evolutionary timeline of the genes involved. Genes that evolved more recently are more likely to be disrupted or deactivated early in cancer progression, while ancient genes, which govern basic cellular functions, become more prominent. Advances in phylostratigraphy, a method that traces the evolutionary origin of genes, seem to support the SAM by allowing researchers to map the chronological sequence of gene evolution, helping to predict the order in which cellular functions are lost or reactivated during cancer progression.

The SAM predicts that the sequence of atavistic reversions should show regularities across different species and cancer types, indicating that despite the diverse origins of cancers, the progression follows a similar pattern due to the underlying evolutionary sequence of gene activation and deactivation. As cancer progresses, cells lose their specialized functions (dedifferentiation), aligning with the concept of atavistic reversions. A key prediction of the SAM is that mutational burden and epigenetic dysregulation will be concentrated in younger genes, which are “less well protected” than the core proliferation pathways, making them more susceptible to dysfunction during carcinogenesis. The SAM thus provides a new appealing framework for understanding cancer as a dynamic process involving multiple evolutionary reversions, where cancer progression goes through a series of atavistic steps.

An alternative view recently proposed for cancer progression posits that neoplastic transformation results from the “gradual uncoupling of the two endosymbiotic subsystems” that are the mitochondria and the nucleus. The former is responsible for the cell bioenergetics, while the latter is responsible for the information preservation. This systemic–evolutionary theory of cancer (SETOC) relies on the dysregulated environment as the causal factor for the uncoupling of the two systems [24,25,26]. The SETOC also defends the atavistic position, since the uncoupling results in the cell regression to more primitive behaviors.

## 4. Where the Metabolism Comes into Play

In the SAM, energy metabolism plays a crucial role, particularly through its connection to the Warburg effect and the metabolic adaptation observed in cancer cells [23]. The Warburg effect (WE), named after the German physiologist Otto Warburg who first described it in the 1920s, refers to the metabolic phenomenon observed in cancer cells characterized by increased glucose uptake and lactate production even in the presence of ample oxygen [18,19]. This results in a shift toward aerobic glycolysis as the primary metabolic pathway for energy production and biomass synthesis. WE contributes to the adaptative resilience of cancer cells by (i) providing rapid ATP generation, (ii) supporting the biosynthesis of macromolecules, and (iii) maintaining the redox state of cells.

WE is one of the most prominent metabolic changes in cancer cells. This metabolic pathway is less efficient than oxidative phosphorylation (OXPHOS) in terms of the amount of ATP produced from 1 mole of glucose, but it supports rapid cell growth and proliferation by providing intermediates necessary for biosynthesis [27]. In the SAM framework, this effect is interpreted as an atavistic reversion to an ancient metabolic state. Early unicellular organisms relied exclusively on glycolysis before the evolution of efficient OXPHOS (Figure 1). By adopting this strategy, cancer cells revert to a more primitive form of energy metabolism that facilitates rapid growth and survival under diverse conditions [28]. This shift to a glycolysis-only metabolism is critical for supporting the increased energy demands of proliferating cells and for the synthesis of nucleotides, amino acids, and lipids required for rapid cell division. This metabolic reversion is driven by both genetic mutations and epigenetic changes that activate ancient pathways. Mutations in key regulatory genes, such as those involved in the PI3K/AKT/mTOR pathway, can upregulate glycolysis and other anabolic processes [29].

The metabolic flexibility conferred by glycolysis provides a significant survival advantage to cancer cells. This allows them to thrive in tumors where the oxygen supply is limited, mimicking, according to the SAM, the metabolic strategies of ancient cells that evolved before the Earth’s atmosphere became oxygen-rich. This metabolic mode supports the high proliferation rate of cancer cells as they progress through the disease, continuously exploiting these primitive metabolic strategies to sustain their growth and division. Within tumors, different cells might exhibit varying degrees of metabolic reversion, reflecting different stages of atavistic progression.

## 5. Critique of the Serial Atavism Model

Lineweaver et al. [23] pushed the serial atavism idea to the epoch before the GOE and proposed among other things, that the cancer cells at a late stage acquire the Warburg phenotype because of the tendency, for eukaryotic cells, to reverse to an evolutionary ancient state when cells lived in total hypoxia. In their words, cancer cells “rely heavily on glycolysis even when oxygen is available. […] When cancer cells prefer glycolysis even when oxygen is available, they are behaving like cells that have reverted to their ancient, glycolysis-only origins. […] SAM hypothesizes that the metabolic shift toward glycolysis during cancer progression is an atavistic reversion”.

The meaning of this sentence is not completely clear, but we interpret it as follows: cancer cells inexorably evolve toward an ancient state corresponding to the pre-GOE period and thus to extreme hypoxia. Cells revert to an upregulated glycolysis state *in the presence of* dioxygen (Warburg effect) because they have an *intrinsic tendency* to go back to an ancient state (which was glycolysis only in an atmosphere deprived of dioxygen). In the SAM, the Warburg effect is thus a direct consequence of the basic SAM hypothesis.

We know little about the ancestral (pre-GOE) energy metabolism, but Lineweaver et al.’s assumption that it was based solely on glycolysis is reasonable, as the ancient cells did not possess a TCA cycle nor an ETC (since mitochondria were not present). On these premises, the authors propose that cancer cells reach a state, in the progression of the disease, with similar features.

The underlying idea of the atavistic hypothesis relies on an overly simplistic (still widely held) view of metabolism, according to which the upregulation of glycolysis is rigidly associated to the inactivation of the TCA cycle and the ETC (since the pyruvate is directed toward lactate and thus does not enter the mitochondria).

We now present some arguments which, we think, invalidate the serial atavistic hypothesis [23], at least in its extension of the pre-GOE period and the metabolism of cancer cells.

(1) *Inconsistency in the order of events*

Lineweaver et al. associated the switch to upregulated glycolysis in the presence of oxygen, known as aerobic glycolysis or Warburg effect (WE), with the evolutionary period before the GOE when oxygen was absent from Earth’s atmosphere. The connection between these two aspects is of course hypoxia. Under hypoxic conditions, healthy mammalian cells normally increase glycolysis and produce lactate. In the context of cancer, the Warburg effect refers to the phenomenon where cancer cells exhibit high glycolytic activity and lactate production even when oxygen is available (supposedly after having been exposed to hypoxia).

The glycolysis-only origin of cells, as mentioned by Lineweaver et al., refers to cells that existed in the pre-GOE world and lacked mitochondria. We consider the emergence of mitochondria via endosymbiosis as a reference point: an event estimated to have occurred around 1.5 to 2 billion years ago [30]. Many aspects of this process remain debated, particularly whether the nucleus evolved due to the presence of mitochondria or if the mitochondrion was the final step in the endosymbiotic process [31]. In any case, according to Lineweaver et al., the evolutionary transition from the absence of oxygen/glycolysis-only (pre-GOE era) to a metabolic regime based on mitochondria (OXPHOS) is an early event, predating the emergence of multicellularity (see their Table 1 and Figure 1). Consequently, the glycolysis-only feature is predicted by the SAM to be one of the latest phenotypic events in cancer progression. In other words, the glycolysis-only phenotype should appear late in oncogenesis in this model.

The timing of when glycolysis becomes dysregulated or upregulated in oncogenesis is a subject of ongoing research and debate, but some evidence suggests that mutation-induced dysregulation in glycolysis can occur early in the process of oncogenesis and possibly even before the development of hypoxic conditions in the tumor microenvironment [29,32]. For example, a recent work [33] investigated how early Warburg metabolism initiates in cancer. Using an inducible zebrafish larval skin pre-neoplastic development model driven by a prevalent oncogenic mutation found in squamous cell carcinoma, namely the human $hRAS^{G12V}$ mutation, the authors explored changes in cellular energy metabolism during the initial stages of tumorigenesis. They induced the $hRAS^{G12V}$ mutation in zebrafish larval keratinocytes to mimic the initial mutational event leading to pre-neoplastic cell (PNC) development. They observed enhanced glycolytic activity in PNCs compared to control cells, as evidenced by higher extracellular acidification rates (ECAR) and the upregulation of key glycolytic enzymes. Additionally, inhibiting glycolysis with low doses of the drug lonidamine (an hexokinase inhibitor) significantly reduced PNC proliferation without affecting cell survival. Their findings indicate that the upregulation of glycolysis is one of the earliest events upon oncogene activation in PNCs. They propose that this could be exploited for PNC eradication. Gatenby and Gillies [32] suggested in 2004 a similar concept, highlighting that the aerobic glycolysis emerges early in the process of carcinogenesis, indicating its potential as a target for cancer prevention.

It should be noted that Lineweaver et al. [23] also considered the possibility that the WE could be associated with a much more recent evolutionary event, given the complex history of rising oxygen levels (Figure 1). Oxygen levels increased significantly during the Neoproterozoic period (0.8–0.5 billion years ago) and later during the Devonian (around 0.4 Gya). This perspective suggests that the WE might in fact be one of the earliest events in oncogenesis. Our conclusion at this point is thus that the SAM has no predictive power regarding the timing of the Warburg effect in the sequence of oncogenic development, and it cannot be falsified on this ground. The next points address other aspects of cancer metabolism which, we think, contradict the SAM.

(2) *Respiration is not systematically impaired*

Otto Warburg published in 1924 in German (and 1930 in English) a theory stating that anaerobic glycolysis was a consequence of some impairment of the respiratory system. He reiterated this theory in 1956 [19], although at that time, it was proven essentially wrong [34]. Experimental works in the following decades confirmed that in a majority of tumors, cells that are in upregulated glycolysis (Warburg) mode also perform respiration (e.g., [34] for a review).

Since then, numerous experimental studies have demonstrated that the TCA cycle and OXPHOS play critical roles in many tumor types. These pathways are now considered potential targets for cancer therapy [35,36]. Here, we highlight only a few experimental results obtained in recent years based on stable-isotope tracing, which is the prevalent method for studying the relative activity of biochemical pathways. The use of ^13^C-glucose to investigate tumor metabolism in vivo began with Fan et al. in 2009 [37]. Their study aimed at examining metabolic alterations in human lung cancer. They found that glycolysis was upregulated compared with non-cancerous tissues from the same patients and that there was also an increased flux through the TCA cycle. More recently, Hensley et al. [38] used ^13^C-glucose and ^13^C-glutamine to map metabolic pathways. They characterized the metabolic heterogeneity in individual lung tumors (NSCLC): some regions relied predominantly on glycolysis, while others showed increased mitochondrial oxidative phosphorylation activity, which was often located in areas with better oxygen supply. For further insights on stable-isotope tracing in the context of tumor metabolism, refer to the review by Bartmann et al. [39].

In other words, in many cancers, respiration is not impaired (contrary to the hypothesis of Warburg), and cancer cells do not obtain their ATP only from glycolysis.

(3) *The Warburg effect is not always a response to hypoxia*

It is generally thought that there is a strong connection between the WE and hypoxia, the common view being that the cells switch from OXPHOS to upregulated glycolysis in hypoxia and remain in this mode even after oxygen is available again. In other words, glycolysis is not inhibited by oxygen. In this section, we aim to show that this is too restrictive.

HIF-1α, the regulatory subunit of HIF-1, is well known as the master regulator of the cellular response to hypoxia. Typically unstable in oxygen-rich conditions, HIF-1α becomes stabilized under hypoxic conditions, enabling it to translocate to the nucleus and form a heterodimer with its binding partner, HIF-1β. This heterodimer then activates the transcription of numerous hypoxia-responsive genes. Under normoxic conditions, HIF-1α is marked for degradation in a process involving prolyl hydroxylase domain (PHD) enzymes and the von Hippel–Lindau protein (VHL). In this process, the HIF prolyl hydroxylases play the role of an oxygen sensor (oxygen being a co-substrate of PHD enzymes).

While hypoxia is a primary inducer of HIF-1α stabilization, various other pathways can also stabilize HIF-1α independent of oxygen levels, which is a condition called pseudohypoxia [40,41]. For example, mutations in the SDHB or SDHD genes, which encode components of succinate dehydrogenase (SDH), also known as respiratory complex II, can lead to an accumulation of succinate. This accumulation inhibits HIF prolyl hydroxylase, resulting in the stabilization of HIF-1α. Other causes of HIF-1α stabilization and accumulation include somatic mutations in the VHL gene, which is common in a majority of patients with a form of kidney cancer called sporadic clear cell renal carcinoma.

Still, other mechanisms operate at the transcriptional and translational levels independently of oxygen levels. For instance, the PI3K/Akt/mTOR and RAS/RAF/MEK/ERK signaling pathways can regulate HIF-1α expression.

Beyond that, it should be noted that aerobic glycolysis can even be exhibited by cells with low HIF levels [32].

(4) *The Warburg effect is reversible*

It has been established that some cancer cells can shift back and forth between glycolytic phenotype and oxidative metabolism in response to factors such as pH, glucose availability, and other environmental conditions. In tumors, lactic acidosis is a common consequence of the cells exhibiting the Warburg effect. Extensive glycolysis produces lactic acid, leading to the acidification of the tumor microenvironment. It has been shown that an acidic pH inhibits glycolysis, whereas lactate itself has a minimal impact on the process [42,43]. This phenomenon parallels observations in muscle tissue: during intense exercise, the pH decreases, inhibiting glycolysis and contributing to muscle fatigue. A key mechanistic aspect of acidosis-induced glycolytic inhibition involves phosphofructokinase-1 (PFK-1), a rate-limiting enzyme in glycolysis and the most important control point in the glycolytic pathway of mammals, the activity of which is reduced under acidic conditions [44].

A recent study [45] quantified the changes in energy metabolism. The authors investigate the metabolic responses of five cancer cell lines, each originating from different tissues, under standard and lactic acidosis conditions. By determining the rates of glucose consumption and lactate production, the glycolysis rate was quantified through the conversion of glucose to lactate. Using isotope lactate tracing, the abundance of glucose-derived lactate was measured, allowing for the calculation of glucose consumption and lactate generation under both conditions. The results indicate that glucose consumption rates under standard conditions were 4.9 to 7.0 times higher than those under lactic acidosis, with HeLa cells exhibiting the lowest rates and AGS cells (gastric cancer cell line) exhibiting the highest rates. Similarly, lactate production rates under standard conditions were 5.1 (HeLa) to 23.8 (AGS) times higher than those under lactic acidosis. Measuring the ATP generation rate from glycolysis and OXPHOS, they found that under lactic acidosis conditions, the oxygen consumption required for ATP synthesis increased significantly by 1.77 to 2.18 times, varying among cell lines. These findings show that under standard culture conditions, glycolysis predominantly contributes to ATP production, while OXPHOS also plays a significant role. Conversely, under lactic acidosis conditions, OXPHOS becomes the primary ATP source. Thus, lactic acidosis induces a metabolic shift in cancer cells from a glycolytic state to an oxidative state.

In another study, Daverio et al. observed [43], on a Human Embryonic Kidney (HEK) cytosolic Cell-Free System, that acidification halted lactate production (the range of pH values investigated was between 7.6 and 6.5).

The reversibility of the WE is most probably one of the causes of metabolic heterogeneity. It is now recognized that tumors consist of a mosaic of cells with distinct metabolic properties. While certain tumor cells rely more on oxygen (and thus on glucose oxidation), others exhibit a predominantly glycolytic metabolism. This metabolic heterogeneity, particularly regarding glucose metabolism within and between human lung tumors, was observed in a study mentioned previously [38].

As a final remark, we would like to emphasize two important points related to this discussion: (*i*) the WE is not permanent and cancer cells do not systematically exhibit it; (*ii*) there is no *discrete switch* between two distinct metabolic regimes. Rather, there is a continuous range of possibilities [46,47]. In addition enhanced glycolysis and oxidative phosphorylation are not mutually exclusive and can be simultaneously upregulated [46].

(5) *The Warburg effect is not exclusive to cancerous cells*

It has been known for a long time that aerobic glycolysis occurs in non-cancerous cells. The first historical example, discovered by Otto Warburg himself, is the mammalian retina [18]. Warburg considered this observation to be an artifact, but it was later proved real [48,49]. This phenomenon is particularly intriguing because retinal neurons are no longer capable of undergoing mitosis. On the application side, since therapies targeting the WE could potentially disrupt retinal metabolism, it is crucial to better understand the molecular mechanisms involved in various cell types [50,51]. Other examples include astrocytes [52,53] and mature erythrocytes. The latter rely on a type of glucose metabolism reminiscent of the WE [54]. Due to the absence of mitochondria, erythrocytes depend almost exclusively on glycolysis to meet their energy requirements.

During early development, embryonic cells display high glycolytic rates. Oginuma et al. [55] demonstrated that chicken embryos and human cells differentiated in vitro from induced pluripotent stem cells exhibit elevated levels of aerobic glycolysis. When embryos were cultured at pH 6.0 or pH 5.3, axis elongation was slowed down and eventualy stalled, which was a behavior similar to the reversible growth arrest observed in cancer cells exposed to a lower intracellular pH. More broadly, rapidly proliferating mammalian cells represent a significant class of non-cancerous cells exhibiting the WE. It includes pluripotent stem cells, immune cells and endothelial cells in angiogenesis and wound repair. Abdel-Haleem et al. [56] suggest that the WE is indeed a “hallmark of rapid proliferation”. Retina cells are a counter-example, since they are not proliferating.

Some immune cells also display aerobic glycolysis upon activation. M1 macrophages predominantly rely on glycolysis in response to tissue injury or infection to support their pro-inflammatory functions. Like cancer cells, M1 macrophages upregulate the glucose transporter GLUT1, increase lactate production, and decrease mitochondrial oxygen consumption [57,58]. Similar metabolic changes have also been observed in activated dendritic cells [59].

The last class encompasses non-cancerous cells within the tumor microenvironment (TME) exhibiting the WE, such as cancer-associated fibroblasts (CAFs) and tumor-associated macrophages (TAMs). The case of CAFs is addressed in the next point.

(6) *Cell cooperation within the TME*

The tumor microenvironment (TME) associated with solid tumors is a highly complex tissue, encompassing a multitude of components, including immune cells and stromal cells. It plays a critical role in tumor initiation and progression. More specifically, the crosstalk between immune cells, stromal cells, and cancer cells regulates the TME. Here, we will only say a few words on the reverse Warburg effect. Pavlides et al. [60] proposed that epithelial cancer cells induce the Warburg effect (aerobic glycolysis) in neighboring stromal fibroblasts. These cancer-associated fibroblasts (CAFs) then undergo myofibroblastic differentiation and secrete lactate and pyruvate. The lactate and pyruvate produced by CAFs can then be taken up by surrounding cancer cells and utilized in the TCA cycle. This effect is termed “reverse” because the non-cancerous cells undergo aerobic glycolysis and supply the cancerous cells with the resulting metabolites. The literature related to the crosstalk between all the cell types in the TME is huge. Our point here is to stress the fact that the microenvironment of tumor cells has nothing in common with the primitive earth environment of pre-GOE cells.

## 6. Discussion

### 6.1. About the Relevance of the SAM as a Conceptual Frame

The SAM proposes that cancer development follows a sequence of reversions, mirroring evolutionary history. On this basis, Lineweaver et al. (2021) [23] proposed a correspondence between specific evolutionary events and events in the course of cancer development. Regarding metabolism, the Warburg effect is viewed as a reversion to a glycolysis-only state, allowing unicellular life in harsh environments. In other words, they establish a parallel between the glycolysis-only life forms in the pre-GOE era and the WE observed in contemporary cells. Due to the extended timeline of atmospheric oxygenation, which spans from approximately 2.4 to 0.4 billion years ago (Figure 1 above), the authors do not provide a definitive prediction regarding the emergence of the corresponding cancer event, which they associate with the WE. However, they lean toward the ’early’ hypothesis, which was around 2 billion years ago (node 41 in their Figure 1, and 6th event in their Table 1), meaning a late occurrence in cancer development. Indeed, the glycolytic genes are probably among the earliest in the history of life, and according to the SAM, the re-emergence of such ancestral traits, here glycolysis, should occur late in cancer progression. This contrasts with the commonly accepted view that places tumor hypoxia as an intermediate event, triggering numerous signals that promote tumor development. The WE provides advantages to cancer cells in the hypoxic and nutrient-deprived tumor microenvironment. In point 1 above, we discussed several studies suggesting that the WE can manifest very early in cancer development. Moreover, it cannot be said that glycolysis is “reawakened”, since it is actually active in all human cells.

The following points, 2 and 3, address important aspects of cancer metabolism: (i) oxidative phosphorylation is not impaired in many cancers and can, in fact, be upregulated; and (ii) the WE is not always a response to hypoxia. These observations are significant because they demonstrate that the WE is not rigidly associated with hypoxia, which is central to Lineweaver et al.’s argument. Additionally, not all cancer cells within a tumor exhibit the WE. A particularly striking work [61] showed that in a colon cancer tumor, glycolysis occurs only in a fraction of cells, and these cells are non-proliferative.

In point 4, we discussed findings indicating that the WE is not permanent but can be reversed depending on the context, allowing cancer cells to exhibit normal levels of glycolysis. Point 5 highlights that non-cancerous cells can also exhibit the WE. Points 2–5 collectively challenge the atavistic reversion hypothesis (i.e., SAM) proposed by Lineweaver et al. [23]. The hypothesis suggests that cancer development mirrors evolutionary events with the WE being a reversion to a primitive, glycolysis-only state. However, the evidence shows that the metabolic behavior of cancer cells is much more flexible and context-dependent than this hypothesis suggests.

Our final point pertains to metabolic cooperation within the tumor microenvironment (TME), particularly through the reverse Warburg effect. However, this specific aspect is just one part of a more complex network of interactions among various cell types in the TME, including immune and stromal cells. These diverse cell types exchange numerous signals, creating an intricate network of interactions. Therefore, the TME can be viewed as an ecosystem where cells are members of a population, and different cell types represent different populations within this ecosystem. In this context, the TME is fundamentally different from the conditions that existed on Earth before the Great Oxygenation Event (GOE). Unlike the pre-GOE environment, which lacked complex multicellular interactions and a stable oxygen supply, the TME is characterized by dynamic cellular communication and adaptation to fluctuating conditions, including oxygen levels. This sophisticated interplay within the TME highlights its mind-boggling complexity and its distinction from the primitive Earth conditions.

### 6.2. Metabolic Plasticity versus Metabolic Rewiring

Points 2 to 5 can be grouped under the concept of metabolic plasticity, which is now recognized as a crucial aspect of tumor development. Metabolic plasticity refers to the ability of cells to shift metabolic fluxes without altering the structure of the metabolic network, meaning no mutations in the enzymes. This adaptability is achieved through regulatory mechanisms at the transcriptional, translational, post-translational and catalytic levels, enabling cells to meet varying physiological demands. In conditions such as tissue repair, immune response, or stem cell differentiation, normal cells can temporarily adopt metabolic profiles similar to those observed in proliferating cancer cells. This flexibility highlights the dynamic nature of cellular metabolism in both normal and cancerous states.

A question in the cancer field is whether gene mutations are mandatory for cancer initiation. These mutations can occur in key regulatory genes involved in metabolic pathways, such as those encoding enzymes of glycolysis, the TCA cycle, and OXPHOS. Oncogenes like MYC and RAS and tumor suppressors like TP53 and PTEN are frequently mutated in cancers and play crucial roles in metabolic adaptation, even metabolic rewiring (also often referred as metabolic reprogramming), if the outcome becomes irreversible. Mutations in these genes can lead to the activation of anabolic pathways and increased glucose uptake, providing the metabolic flexibility necessary for rapid cell division and tumor growth. This gene-centered view suggests that the metabolic phenotype of cancer cells is a product of evolutionary selection at the cell population level, where mutations that confer a growth advantage accumulate.

It seems likely that the supportive metabolism in cancer may result from a combination of metabolic plasticity (no mutations) and metabolic rewiring (mutations). Metabolic plasticity allows normal cells to adapt quickly to proliferative signals, while the accumulation of gene mutations during tumorigenesis locks cells into a state of perpetual growth and division. This is in relation with the ongoing debate between the Somatic Mutation Theory (SMT) and the Tissue Organization Field Theory (TOFT) [62,63,64] each pertaining to one aspect of the metabolic adaptation. The first refers to early mutations as the trigger of cancer, while the second puts forward cell plasticity under environmental pressure.

## 7. Conclusions

The SAM suggests that cancer progression mirrors evolutionary stages toward more and more primitive states, the Warburg effect corresponding to ancient unicellular cells in the pre-GOE era. Although intuitively appealing, it is not an adequate conceptual frame. It is too narrow and simplistic, failing to account for the metabolic plasticity of cancer cells and the critical role of the TME. Cancer metabolism is highly adaptable, altered by genetic and environmental factors, and it dynamically interacts with the TME. While the SAM falls short, the general idea of viewing cancer through an evolutionary lens retains value [65,66].

## Data Availability

Not applicable.

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
