# Peer review of "Is Cancer Metabolism an Atavism?"

_cancers, 2024, doi:10.3390/cancers16132415_

Round 1

Reviewer 1 Report

Comments and Suggestions for Authors

The atavistic theory formulated by Davies and Lineweaver and Vincent proposes that the origin and development of cancer is explained by the regression of cellular phenotypes to primitive types. The atavistic theory assumes that the cancer cell reactivates and expresses an ancestral program buried in its genome and inherited from the preproterozoic era. In the manuscript The serial atavism model/ atavistic reversion hypothesis proposed by Lineweaver and colleagues in 2021, is questioned.

This overview article is well written. The reader is offered a systematic presentation of information. The narrative is fluid and engaging. Explanation and conclusion are appropriate. However, I have concerns about the accuracy, credibility, and appropriate interpretation of the information provided, particularly in paragraph (2) Respiration is not systematically impaired, which concludes that »in many cancers, respiration is not impaired (contrary to Warburg's hypothesis) and cancer cells do not derive their ATP solely from glycolysis.

The authors should more specifically explain whether mitochondrial OxPhos is normal (or not) in most cancer cells by not ignoring the massive evidence of abnormalities in mitochondrial number, structure and function found in all major cancers (e.g. see evidence in paper Seyfried TN, Arismendi-Morillo G, Mukherjee P, Chinopoulos C. On the Origin of ATP Synthesis in Cancer. iScience. 2020 Nov 2;23(11):101761. doi: 10.1016/j.isci.2020.101761.).  Abnormalities in the number of mitochondria, structural abnormalities in the mitochondrial cristae, changes in the mitochondrial lipids and enzymes of the electron transport chain as well as abnormalities in the mitochondrial membranes have been documented in all major types of cancer. The fact that structure determines function is a fundamental principle of evolutionary biology. Besides, it is important to note that much of the evidence supporting the view that OxPhos is not seriously affected in cancer cells comes from in vitro studies (see references in paper Seyfried TN, Arismendi-Morillo G, Mukherjee P, Chinopoulos C. On the Origin of ATP Synthesis in Cancer. iScience. 2020 Nov 2;23(11):101761. doi: 10.1016/j.isci.2020.101761). It should also be noted that aerobic fermentation (Warburg effect) is not involved in rapid cell proliferation during liver regeneration or intestinal cell regeneration in vivo. ATP synthesis in these processes is carried out by OxPhos. Glucose indeed inhibits the regeneration of liver cells. The information on liver regeneration does not support the hypothesis that aerobic fermentation is common to all proliferating cells. It should be noted that the Crabtree effect can explain aerobic fermentation in normal cells that proliferate in vitro. In other words, aerobic fermentation is an artifact of the in vitro environment.  Cancer cells, on the other hand, use aerobic fermentation whether grown in vivo or in vitro.  They do this because their mitochondria are defective, as Warburg first proposed.  

In order to avoid a one-sided presentation of facts and possible distortions (e.g. point 2: oxidative phosphorylation is not impaired in many types of cancer), I recommend that the authors also read the articles by Thomas N. Seyfried et al. and include their results in the manuscript in a meaningful way or refute them with arguments.

Reviewer 2 Report

Comments and Suggestions for Authors

Dear authors,

Congratulation for your work. It is indeed of great scientific value since it sheds a new light on such complex issues as cancer development and cancer metabolism. Also, I appreciate the complex questions that you have highlighted that renders SAM an incomplete model. The parer is well written, easy readable (inspite of the hard topic).

One remark though: you have a repeating paragraph in between lines 397-399 and 400-402. Please delete one of the two.
